# Anti-Obesity Effect of Different *Opuntia stricta* var. *dillenii*’s Prickly Pear Tissues and Industrial By-Product Extracts in 3T3-L1 Mature Adipocytes

**DOI:** 10.3390/nu16040499

**Published:** 2024-02-09

**Authors:** Iván Gómez-López, Itziar Eseberri, M. Pilar Cano, María P. Portillo

**Affiliations:** 1Laboratory of Phytochemistry and Plant Food Functionality, Biotechnology and Food Microbiology Department, Institute of Food Science Research (CIAL) (CSIC-UAM), Nicolás Cabrera 9, 28049 Madrid, Spain; ivan.gomez@ehu.eus (I.G.-L.); mpilar.cano@csic.es (M.P.C.); 2Nutrition and Obesity Group, Department of Nutrition and Food Science, Faculty of Pharmacy and Lucio Lascaray Research Center, University of the Basque Country (UPV/EHU), 01006 Vitoria-Gasteiz, Spain; itziar.eseberri@ehu.eus; 3CIBERobn Physiopathology of Obesity and Nutrition, Institute of Health Carlos III (ISCIII), 01006 Vitoria-Gasteiz, Spain; 4BIOARABA Institute of Health, 01006 Vitoria-Gasteiz, Spain

**Keywords:** *Opuntia stricta* var. *dillenii*, betalains, phenolic compounds, anti-obesity effect in 3T3-L1, mechanism involved

## Abstract

*Opuntia stricta* var. *dillenii* fruit is a source of phytochemicals, such as betalains and phenolic compounds, which may play essential roles in health promotion. The aim of this research was to study the triglyceride-lowering effect of green extracts, obtained from *Opuntia stricta* var. *dillenii* fruit (whole fruit, pulp, peel, and industrial by-products (bagasse)) in 3T3-L1 mature adipocytes. The cells were treated on day 12, for 24 h, after the induction of differentiation with the extracts, at doses of 10, 25, 50, or 100 μg/mL. The expression of genes (PCR-RT) and proteins (Western blot) involved in fatty acid synthesis, fatty acid uptake, triglyceride assembly, and triglyceride mobilisation was determined. The fruit pulp extraction yielded the highest levels of betalains, whereas the peel displayed the greatest concentration of phenolic compounds. The extracts from whole fruit, peel and pulp were effective in reducing triglyceride accumulation at doses of 50 μg/mL or higher. Bagasse did not show this effect. The main mechanisms of action underpinning this outcome encompass a reduction in fatty acids synthesis (de novo lipogenesis), thus limiting their availability for triglyceride formation, alongside an increase in triglyceride mobilisation. However, their reliance is contingent upon the specific *Opuntia* extract.

## 1. Introduction

Obesity is defined as excessive fat accumulation in adipose tissue, which is closely related to the incidence of non-communicable diseases, such as cardiovascular diseases, type-2 diabetes, and some cancers, among others [1]. The prevalence of obesity is increasing, and nowadays, this condition is one of the leading causes of premature mortality and disability worldwide [2]. Given its widespread occurrence, the prevention and treatment of obesity emerge as leading priorities for health systems. The interventions for obesity constitute lifelong therapeutic measures, and their efficacy is frequently diminished due to patients’ poor adherence to diet and physical activity programs [3].

The association between diet and health has been demonstrated throughout the course of human history [4]. Numerous studies have shown that bioactive compounds naturally present in some food stuffs and plants act as important modulators in the prevention of obesity [5,6,7,8]. In this context, the genus *Opuntia* spp., belonging to the *Cactaceae* family, represents a group of plants of great interest for several reasons. On the one hand, these plants are a great source of numerous compounds, such as polyphenols, betalains and pectins, among others, that can induce beneficial effects on human health [9,10,11,12]. On the other hand, climate change is causing global warming and drought, and in this scenario, *Opuntia* spp. shows a great adaptability to arid and semi-arid environments, demonstrating their capacity to produce fruit in these conditions [13,14].

Currently, over 300 species of *Opuntia* spp. have been identified [15]. Although *Opuntia ficus-indica* (*O. ficus-indica*) is the most commercialised, consumed, and investigated species, there are others that grow wildly, such as *Opuntia stricta* var. *dillenii* (*O. stricta* var. *dillenii*) [16]. An interesting feature of these plants is their fruit, a small dark purple haw commonly referred to as the prickly pear. One of the main groups of bioactive compounds in *O. stricta* var. *dillenii* prickly pears are betalains, nitrogen-based pigments synthesised as secondary metabolites and accumulated in vacuoles in the plant cell cytoplasm. There are two different groups: betaxanthins (yellow-orange dye) and betacyanins (reddish-purple dye) [17]. The intense purple colour of the *O. stricta* var. *dillenii* fruits signifies their abundance in betacyanins, especially in betanin, isobetanin, and neobetanin [18,19]. Additionally, this cactus fruit contains a substantial quantity of phenolic compounds, primarily composed of phenolic acids, notably piscidic acid and flavonoids, particularly isorhamnetin. Isorhamnetin glucosyl-rhamnosyl-pentoside (IG2) emerges as the most abundant flavonoid, along with the presence of quercetin [18,20,21]. These bioactive compounds enhance the appearance and flavour of the fruit, while offering potential human health benefits, such as antioxidant and anti-inflammatory effects [22,23,24,25].

The aim of this research was to study the triglyceride-lowering effect of extracts rich in betalains and phenolic compounds derived from various tissues of *O. stricta* var. *dillenii*’s (including whole fruit, pulp, and peel) and an industrial by-product (bagasse) in 3T3-L1 mature adipocytes in order to determine the most suitable starting material to obtain an extract with proven biological activities. The potential mechanisms involved in this effect were also analysed.

## 2. Materials and Methods

### 2.1. Plant Material

*O. stricta* var. *dillenii* prickly pears were collected in September 2020 in Tinajo, Tenerife, The Canary Islands, Spain (28°2′ N, 16°4′ W at 209 m above sea level). Fruits were cleansed and chosen based on their ripeness, size, and colour, with damaged ones being rejected. The fruits were then processed manually into peels, pulps, and whole fruit tissues (Figure 1). Additionally, Bernardo’s company (Lanzarote, Spain) supplied the by-product of *O. stricta* var. *dillenii* prickly pear-referred to as bagasse—obtained from the jam industry. All these samples were sliced into small cubes (1 × 1 cm), immediately frozen with liquid nitrogen (N_2_), and subsequently freeze-dried. Freeze-dried tissues were pulverised to a fine particle size (<2 mm), removing the seed. Freeze-dried *O. stricta* var. *dillenii* whole fruit, peel, pulp, and bagasse powders were stored at −24 °C and packaged in vacuum-sealed bags to avoid exposure to oxygen.

### 2.2. Opuntia stricta var. dillenii Extracts

Extracts rich in betalains and phenolic compounds were obtained from freeze-dried samples of tissue under reduced light conditions, as previously documented by Gómez-López et al. [18]. One gram of each freeze-dried tissue from *O. stricta* var. *dillenii* was subjected to extraction using 5 mL of 50% ethanol in water (1:1, *v*/*v*). This process was iterated twice more with 3 mL of 50% ethanol in water (1:1, *v*/*v*). The last extraction was conducted using 3 mL of 100% ethanol. The solvents in the supernatants were removed using a rotary evaporator (Buchi, Flawil, Switzerland) at 25 °C to a minimum volume (5 mL). Aliquots of each tissue extract were freeze-dried and stored at −20 °C until their use.

### 2.3. Characterisation of Opuntia Extracts

#### 2.3.1. HPLC Analysis of Betalain and Phenolic Compounds

The analyses of betalains and phenolic compounds in various tissue extracts of *O. stricta* var. *dillenii* were performed simultaneously using high-performance liquid chromatography with diode-array detection (HPLC-DAD), following the methodology previously reported by our research group [18,26]. In summary, a 1200 Series Agilent HPLC System (Agilent Technologies, Santa Clara, CA, USA) equipped with a C18 reverse column (Zorbax SB-C18, 250 × 4.6 mm i.d., S-5 μm; Agilent) at 25 °C were used. A gradient elution over 70 min was carried out using Phase A, comprising ultrapure water with 1% formic acid (*v*/*v*), and Phase B, consisting of methanol (99.8% LC-MS) with 1% formic acid (*v*/*v*). The injection volume selected for the analysis was 20 μL, and the flow rate was determined at 0.8 mL/min. To simultaneously monitor various chemical compound families, the UV-visible photodiode array detector was configured to operate at four wavelengths. According to the absorption bands, phenolic acids were detected at 280 nm and flavonoids at 370 nm [27]. The absorbance for the reddish-purple betacyanins was detected at 535 nm, whereas for the yellow betaxanthins, it was recorded at 480 nm [28]. To validate the chemical composition of each bioactive compound, additional analyses were performed using an LCMS SQ 6120 electrospray ionization (ESI) mass spectrometry detector (Agilent Technologies, Santa Clara, CA, USA) in positive ion mode. The drying gas was nitrogen at 1.379 bar, and 50 mL/s were used. The capillary had a potential of 3.5 kV, and the nebulizer temperature was 300 °C. Helium was the coliseum gas, and at 70 V the fragmentation amplitude was performed. From 100 to 1000 *m*/*z*, the spectra were recorded. Moreover, the characterisation and mass spectrometry analyses were conducted using MaXis II LC-QTOF equipment (Bruker Daltonics, Bremen, Germany) which was equipped with an ESI source and maintained the same chromatographic configuration.

Compounds were identified based on their retention times (rt), UV/Vis maximum absorption (λmax), and mass spectral (*m*/*z*) data, comparing them to commercial, semi-synthesized, or purified standards. Quantification of the most abundant beta-lains, piscidic acid, and isorhamnetin glycosides was performed using calibration curves derived from corresponding isolated standards. Betanin was purified using Sephadex L20 resin from a commercial betalin-rich concentrate extract of beetroot, and betaxanthins were semi-synthesized from pure betanin following procedures outlined by García-Cayuela et al. [26]. Phyllocactin was extracted from cactus berry fruits (*Myrtillocactus geometrizans*), and piscidic acid was isolated from prickly pear peels using semi-preparative high-performance liquid chromatography (HPLC), as detailed by Montiel-Sánchez et al. [29] and García-Cayuela et al. [26]. Isorhamnetin glycoside standards were provided by Dr. Serna-Saldivar’s laboratory at the Biotechnology Center FEMSA (Instituto Tecnológico de Monterrey, Monterrey, Mexico).

The complete description of UV-vis and mass spectroscopy characteristics of all individual betalains and phenolic compounds found in *O. stricta* var. *dillenii* extracts were previously reported by Gómez-López et al. [18]. Appendix A present the descriptive data and chromatograms obtained from HPLC-DAD for the most abundantly identified compounds in whole fruit (WF), peel (PE), pulp (PU), and bagasse (BA) extracts from the *O. stricta* var. *dillenii*.

#### 2.3.2. Determination of the Chemical In Vitro Antioxidant Activity Using ORAC Method and LOX-FL

Antioxidant activity was assessed using two different methods: the oxygen radical absorbance (ORAC method), determined using fluorescence degradation, and the sensibility of the lipoxygenase fluorescein (LOX-FL method). The ORAC antioxidant activity was assessed according to the reported method by Gómez-López et al. [30]. The assay was carried out in an automated plate fluorimeter reader (Bio-Tek Instruments Inc., Winooski, VT, USA) in a 96-well microplate, where 20 μL of diluted extracts in PBS (phosphate-buffered saline) at 0.075 M and pH 7.4 were mixed with 120 μL of 11.7 μM fluorescein disodium. Then, samples were incubated for ten minutes at 37 °C. To generate peroxyl radicals, 60 μL of AAPH (2,2′-Azabis(2-methylpropionamidine) dihydrochloride) at 0.153 M were added. In the microplate reader, 91 measures were recorded, one per minute (for a total of 91 min), at 485 nm the excitation condition and at 530 nm the emission condition. A Trolox calibration curve was employed for quantification that ranged from 0.01 to 0.045 μmol Trolox/mL. The results were expressed as μmol Trolox equivalents/g dry weight (DW).

LOX-FL antioxidant activity was measured using the method previously outlined by Gomez-Maqueo et al. [31]. The LOX-FL technique was executed using a spectrophotometer (Specord 210 Plus, Analytik Jena, Jena, Germany), with the reaction monitored at 485 nm. The reaction mixture (1 mL) contained 0.1 M sodium-borate buffer (pH 9.0), 400 μM sodium-linoleate, 1 μL Tween 20 per μmol linoleate, and 4.5 μM fluorescein. A 0.5 EU of soybean lipoxygenase was added to start the response. In brief, the methodology involved the inhibition of the LOX-FL reaction by calculating the decrease in the rate of fluorescein bleaching in the presence of extracts (Va) relative to the control (Vc) using the following equation:Inhibition (%) = [1 − (Va/Vc)] × 100(1)

The antioxidant capacity of the extracts was calculated using a dose-response curve obtained with Trolox, and the results were expressed as μmol Trolox equivalents/g DW.

### 2.4. Mature Adipocyte Cell Experimental Design

3T3-L1 pre-adipocytes, supplied by American Type Culture Collection (Manassas, VA, USA), were cultured in DMEM with 10% FBS (foetal bovine serum). The medium was changed every two days until cell harvesting on day 12, at which point over 90% of the cells had reached maturity as adipocytes with visible lipid droplets. Following confluence at day 0, cells were exposed to DMEM with 10% FBS, 10 g/mL insulin, 0.5 mM IBMX (isobutylmethylxanthine), and 1 mM DEXA (dexamethasone) for two days to induce differentiation. In the subsequent two days, the differentiation medium was replaced with FBS/DMEM (10%) supplemented with 10 g/mL insulin. Commencing from day four onward, the differentiation medium was substituted with FBS/DMEM (10%) enriched with 0.2 g/mL insulin. All media contained 1% penicillin/streptomycin (10.000 U/mL). Cells were maintained at 37 °C in a humidified CO_2_ atmosphere (5%).

#### 2.4.1. Cell Treatment

For mature adipocyte treatment, cells grown in 6-well plates were incubated with whole fruit, peel, pulp, or bagasse (BA) extracts from *O. stricta* var. *dillenii* at 10, 25, 50, or 100 µg/mL (diluted in milli-Q water), on day 12 after differentiation, for 24 h. Subsequently, cells were collected for triglyceride and protein determination, and RNA extraction. Each experiment was performed in triplicate.

#### 2.4.2. Cell Viability Assay

Cells were seeded onto 96-well plates and were maintained and subjected to treatment under the same conditions as previously outlined, using *O. stricta* var. *dillenii* aqueous extracts at 10, 25, 50, 100 µg/mL. Cell viability was assessed through crystal violet staining for living cells, following the protocol described by Gilles et al. [32]. In summary, cells were rinsed with PBS, fixed using 3.7% formaldehyde, and then stained with crystal violet (0.25%) in the dark for 30 min. The formed crystals were then dissolved in acetic acid (33%) and the absorbance was measured at 590 nm using an iMark microplate reader (Bio-Rad, Hercules, CA, USA). The absorbance of each well was proportional to cell density. Results are expressed in arbitrary units.

#### 2.4.3. Measurement of Triglyceride and Protein Content in 3T3-L1 Mature Adipocytes

Following cell treatment, the medium was removed, and mature adipocytes were thoroughly washed with phosphate-buffered saline (PBS) and harvested using a 300 µL of buffer comprising Tris-HCl at pH 7.4, 0.15 M sodium chloride (NaCl) and 1 mM ethylene diamine tetra acetate (EDTA) with protease inhibitors (0.1 M phenylmethylsulphonyl fluoride and 0.1 M iodoacetamide). Afterwards, cell samples were disrupted using ultrasound equipment, the Branson Digital Sonifier SFX 550 (Emerson Electric Co, St. Louis, MO, USA) with a 2 mm diameter ultrasound-microprobe (Biogen Scientific S.L., Madrid, Spain). Triglyceride content (mg/mL) was measured using Infinity Triglycerides reagent (Thermo Scientific, Rockford, IL, USA). The lipid content of cells was standardised by the protein content of each well. Protein content was determined using the method described by Bradford et al. [33]. Results are expressed as mg of triglycerides/mg of protein, and presented in arbitrary units.

#### 2.4.4. RNA Extraction and RT-PCR

Trizol (Invitrogen, Carlsbad, CA, USA) was used to extract the RNA from the mature adipocytes. An RNA 6000 Nano Assay (Thermo Scientific, Wilmington, DE, USA) was used to confirm and quantify the integrity of the RNA extracted from all samples. To remove any possible genomic DNA contamination, RNA samples were processed with a DNase I kit (Applied Biosystems, Foster City, CA, USA). A measure of 1.5 μg of total RNA of each sample underwent reverse transcription to first-strand complementary DNA (cDNA) using the iScript cDNA Synthesis Kit (Bio-Rad, Hercules, CA, USA). The reactions involved an initial incubation for ten minutes at 25 °C, followed by 120 min at 37 °C, and concluded with a five-minute incubation at 85 °C.

The mRNA levels of relative acetyl CoA carboxylase (*acc*), fatty acid synthetase (*fas*), cluster of differentiation 36 (*cd36*), lipoprotein lipase (*lpl*), Diacylglycerol O-Acyltransferase 2 (*dgat2*), adipose triglyceride lipase (*atgl*), and hormone-sensitive lipase (*hsl*) were measured in mature adipocytes using real-time PCR. The quantification was performed using an iCycler-MyiQ Real-Time PCR Detection System (BioRad, Hercules, CA, USA). *β*-actin was used as the housekeeping (reference gene). The PCR reagent mixture consisted of 4.75 μL aliquot of each diluted cDNA, SYBR Green Master Mix (Applied Biosystems, Foster City, CA, USA), along with forward and reverse primers at a concentration of 5 nM. For *fas*, *cd36*, *atgl* and *lpl*, the reagent mixture consisted of 4.5 μL of each cDNA, Premix Ex Taq TM (Takara, San Jose, CA, USA), and the upstream and downstream primers. Table 1 shows the sequences and PCR conditions of the specific commercially synthesised primers. All mRNA levels were normalised to the values of *β*-actin, and the results were expressed as fold changes in threshold cycle (Ct) values relative to the controls using the 2^−ΔΔCt^ method [34].

#### 2.4.5. Western Blot Analysis

Acetyl CoA carboxylase (ACC), glucose transporter type 4 (GLUT4), and hormone-sensitive lipase (HSL) protein quantification was carried out using αTubulin as housekeeping. In order to determinate the activity of ACC and HSL, phosphorylated Acetyl CoA carboxylase (Phospho-ACC) and phosphorylated hormone-sensitive lipase (HSL) were measured.

Immunoblot analyses were conducted using 15 μg of protein extracted from mature adipocytes. The separation of proteins was achieved through electrophoresis in precast 4–15% sodium dodecyl sulphate (SDS)-polyacrylamide gradient gels (Bio-Rad, Hercules, CA, USA), followed by transfer to PVDF (polyvinylidene difluoride) membranes (Merck, Darmstadt, Germany). Membranes were blocked for two hours at room temperature using a casein PBS-Tween solution at 5%. After that, they were blotted overnight at 4 °C using the appropriate antibodies. The protein levels were detected via specific antibodies for phosphorylated ACC (Serenine 79, 1:1000), phosphorylated HSL (Serenine 660, 1:1000) (Cell Signaling Technology, Danvers, MA, USA) and GLUT4 (1:1000) (Santa Cruz Biotech, Dallas, TX, USA). Afterward membranes were incubated with polyclonal anti-rabbit (1:5000) (Santa Cruz Biotech, Dallas, TX, USA) for two hours at room temperature, and proteins levels were measured. After antibody stripping, the membranes were blocked and subsequently incubated with ACC (1:1000), HLS (1:100) (Cell Signaling Technology, Danvers, MA, USA) and tubulin (1:1000) (Santa Cruz Biotech, Dallas, TX, USA) antibodies, repeating the process. Employing an ECL system (Thermo Fisher Scientific Inc., Rockford, IL, USA), the bound antibodies were detected, and using a ChemiDoc MP Imaging System (Bio-Rad, Hercules, CA, USA), they were quantified. Specific bands were identified with a standard loading buffer (Precision Plus protein standards dual-color; Ref. 161-0374, Bio-Rad).

### 2.5. Statistical Analysis

Data were analysed using SPSS Statistic software 26.0 (IBM corp., Armonk, NY, USA) and expressed as mean ± standard error of the means. Results were evaluated through one-way analysis of variance (ANOVA) followed by *post hoc* Tukey’s-b test. For assessment related to cell viability, triglyceride-lowering effects, as well as gene and protein expressions, no comparisons among all the cells groups were carried out. Instead, cells treated with each extract were compared with the control cells using Student’s *t* test. Statistical significance was set at the *p* ≤ 0.05 level.

## 3. Results and Discussion

### 3.1. Opuntia stricta var. dillenii Extracts Characterisation and In Vitro Biological Activities

#### 3.1.1. Bioactive Compounds from *Opuntia stricta* var. *dillenii* Extracts: Betalains and Phenolic Compounds

The identification and quantification data of the primary bioactive compounds in the extracts are presented in Table 2. This analysis was conducted according to the retention time, UV/V is maximum absorption (λmax), and mass spectra data (*m*/*z*), and it is guided using methodologies outlined in a recent publication [18]. The HPLC-DAD chromatograms of the main compounds present in the extracts are shown in the Appendix A.

##### Betalains

Due to their dark-purple colour, the main compounds among betalains in these fruits are betacyanins [18]. The most representative compound within betacyanins was betanin (peak 2), which was present in a range of 2.99 to 2.91 mg/g DW in *O. stricta* var. *dillenii*, depending on the fruit tissue extracts, and 0.84 ± 0.02 mg/g DW in the bagasse (industrial by-product) extract. Nonetheless, the *O. stricta* var. *dillenii* fruits were also rich sources of isobetanin (Peak 3), 2′-*O*-apiosyl-4-*O*-phyllocactin (Peak 6), 5″-*O*-E-sinapoyl-2′-apyosil-phyllocactin (Peak 7), and neobetanin (Peak 8). A consistent betalain pattern was evident across *O. stricta* var. *dillenii*’s prickly pear, wherein the levels of these pigments were consistently higher in fruit tissues compared to the bagasse. However, the peel extract exhibited the lowest neoteanin content (Peak 8), registering at 0.82 ± 0.00 mg/g DW.

*O. stricta* var. *dillenii* pulp extract displayed the highest total betalain content (12.78 ± 0.48 mg/g DW), with neobetanin (peak 8) being the most abundant, with 3.26 ± 0.48 mg/g DW. In contrast, the bagasse extract accounted for the lowest amount (3.24 ± 0.27 mg/g DW). The bagasse contained compounds in their descarboxyl form, arising from the degradation of betalains [18], therefore, showing a low content of betacyanins. In accordance with the study reported by Gómez-Maqueo et al. [35], which focused on the dynamic microstructural analysis of prickly pear cells, it was determined that betalains are localised in the vesicles (in the cell wall) and vacuoles (in the cytoplasm) of the prickly pear cells. In comparison to different O. *stricta* var. *dilenii* tissues, several authors have consistently asserted that the pulp stands out as the prickly pear tissue with the highest concentration of betalains [18].

##### Phenolic Compounds

Regarding the phenolic compounds, the *O. stricta* var. *dillenii*’s fruits were found to be rich in phenolic acids and flavonoids [18]. Among phenolic acids, piscidic acid (peak 1) emerged as the predominant compound, ranging from 2.33 ± 0.33 mg/g DW in the peel extract to 0.62 ± 0.05 mg/g DW in the pulp extract. These findings align well with a study previously reported by our group, where *O. stricta* var. *dillenii* peel was the fruit tissue richest in piscidic acid [18]. In the case of flavonoids, isorhamnetin-glucosyl-rhamnosyl-pentoside (IG2) (peak 13) was the most abundant, with quantities ranging from 0.52 ± 0.02 mg/g DW in the peel extract (PE) to 0.05 ± 0.00 mg/g DW in the pulp extract. The peel extract exhibited the highest concentration of total major phenolic compounds, represented as the sum of individually identified phenolic compounds, at 3.04 ± 0.02 mg/g DW; the pulp extract showed the lowest quantity, at 0.67 ± 0.05 mg/g DW (Table 2). These results are consistent with the findings reported by other authors, indicating that the peel tissue of the prickly pear is richer in phenolic compounds than the pulp tissue [26].

#### 3.1.2. In Vitro Biological Activities of *Opuntia stricta* var. *dillenii* Extracts

Oxidative stress plays an important role in the pathogenesis of obesity and its co-morbidities [36]. Consequently, analysing the antioxidative activity of a potential anti-obesity tool is a matter of interest.

The antioxidant activity of the *O. stricta* var. *dillenii* extracts was measured. The ORAC assay stands out as the most conventional method; nevertheless, Gómez-Maqueo et al. [31] proposed the LOX-FL method as the optimal choice for in vitro studies. This recommendation is based on its ability to better mirror the in vivo antioxidant capacity of betalains, substantiated by the correlation between LOX-FL assay and betalain content. Soccio et al. [37] also recommended the LOX-FL method to determine the antioxidant activity of a large variety of plant foods, which possesses a high-level ability to discriminate among different samples. Taking this into account, both methods were used in the present study. Both determinations showed that the four analysed extracts (whole fruit, peel, pulp, and bagasse from *O. stricta* var. *dillenii*) displayed antioxidant activity (Figure 2 and Appendix A).

The present results confirm the established antioxidant properties of betalains [38,39] and phenolic compounds [40]. Comparison of the extracts of *O. stricta* var. *dillenii* with those from other *Opuntia* spp. indicates a superior antioxidant activity in the former [41,42]. This antioxidant activity could be related to health-promoting properties. Indeed, an in vivo study correlated the antioxidant and anti-inflammatory effects of *O. ficus-indica* cladode extract to its beneficial impact on human health and metabolic diseases in supplemented pasta (3%) [8]. Furthermore, a review on oxidative stress, plant natural antioxidants, and obesity has determined that natural antioxidants play crucial roles in managing obesity [43].

### 3.2. Effects of Opuntia stricta var. dillenii Extracts in 3T3-L1 Mature Adipocytes

#### 3.2.1. Effects on Cell Viability

3T3-L1 mature adipocytes were incubated with 10, 25, 50, or 100 µg/mL of *O. stricta* var. *dillenii* extracts for 24 h. Figure 3 shows a lack of significant differences, in terms of cell viability between treated cells and the controls, meaning that the extracts were not cytotoxic. Consequently, if any reduction in adipocyte triglyceride content effect is shown, it should be attributed to the triglyceride-lowering effect of the extracts.

#### 3.2.2. Effects on Triglyceride Content

The lipid quantification assay substantiated a significant decrease (*p <* 0.05) in intracellular triglyceride levels in mature 3T3-L1 adipocytes treated with 50 µg/mL or 100 µg/mL of the whole fruit, peel, and pulp extracts from *O. stricta* var. *dillenii* prickly pear. By contrast, bagasse extracts did not show a significant reduction with respect to the non-treated cells (controls).

When cells were treated with *O. stricta* var. *dillenii* whole fruit extract at 50 µg/mL or 100 µg/mL, the triglyceride content decreased by 24.7% and 34.6%, respectively. The peel extract, at the same doses, induced a 23.6% and 21.9% reduction, respectively (Figure 4 and Appendix A). Adipocytes treated with the pulp extract at 50 µg/mL or 100 µg/mL showed the highest reductions in triglyceride content (−37.4% and −34.2%). These data reveal that, at a dose of 50 µg/mL, pulp extract was the most effective. By contrast, at a dose of 100 µg/mL, the efficacy of both whole fruit and pulp were similar. Altogether, these results allow us to conclude that among the four *O. stricta* var. *dillenii* extracts analysed, the pulp extract is the most interesting because it exerted the greatest delipidating effect at the lowest dose. Moreover, as it can be observed in Figure 4, there is no discernible dose-response pattern in the triglyceride-lowering effect, at least within the concentration range of 50–100 µg/mL. To the best of our knowledge, this is the first instance that *O. stricta* var. *dillenii* extracts have been evaluated in 3T3-L1 mature adipocytes. Moreover, no prior studies have been reported to test the impact of extracts obtained from other *Opuntia* spp. on triglyceride accumulation in adipocytes.

Other authors have incubated adipocytes with extracts derived from Pitaya fruit (*Hylocereus polyrhizus* cv. *Jindu*), a plant also belonging to the *Cactaceae* family, which is rich in betalains, specifically betanins, with concentrations of 7.44 ± 0.01 mg/g in the pulp and 9.44 ± 0.01 mg/g in the peel. These studies have also found significant lipid-lowering effects, with the peel being the most effective. The authors declared that the lipid-reducing effect of pitaya peel could be attributed to its betacyanin content (mainly betanin) [44].

According to the extract composition (Table 2), the pulp extract, previously identified in this section as the most effective, exhibits the highest content of betalains, particularly neobetanin, and the lowest content in phenolic compounds. This observation, consistent with the previously mentioned results, implies that betalains may be the primary contributors to this biological effect. Nevertheless, additional investigation is required to substantiate this hypothesis.

#### 3.2.3. Effects on Genes and Proteins Involved in 3T3-L1 Mature Adipocytes Metabolism

In view of the reduction in triglyceride content induced by the *O. stricta* var. *dillenii* extracts in mature adipocytes, gene and protein expression were assessed in order to understand the mechanisms underlying the observed effects. For this purpose, cells treated with the lowest effective dose of the whole fruit, the peel, and the pulp extracts (50 µg/mL) were used. The genes and proteins selected were involved in fatty acid synthesis, fatty acid uptake, triglyceride assembly, and triglyceride mobilisation.

Concerning triglyceride synthesis, mature adipocytes rely on glucose to obtain glycerol-phosphate, which will be combined with fatty acids to synthesise triglycerides. The influx of glucose into mature adipocytes is mediated by the glucose transporter type 4 (GLUT4), which is up-regulated by insulin. In 3T3-L1 mature adipocytes treated with *O. stricta* var. *dillenii* extracts, GLUT4 protein expression was not modified (Figure 5). Regarding de novo lipogenesis, the process that provides fatty acids by synthesising them from Acetyl-CoA, *acc* expression was not significantly modified by any of the treatments, and *fas* expression was only reduced (*p <* 0.05) in cells that were incubated with the pulp extract (Figure 6). As ACC is the limiting enzyme in de novo lipogenesis, the enzyme activity was measured using the ACC-phosphorylated/total-ACC ratio, in order to identify possible modifications at a posttranscriptional level. All extracts significantly increased (*p <* 0.05) this ratio (Figure 5). Taking into account that the phosphorylated form of this enzyme is the inactive one, the extract treatments reduced the enzyme activity.

The gene expression of the transmembrane protein *cd36*, responsible for the uptake of fatty acids from the blood, was only reduced in adipocytes treated with the pulp extract. These results indicate that only in the case of this extract, a reduced availability of fatty acids can be involved in the triglyceride reduction observed in adipocytes. With regard to gene expression of *lpl*, the enzyme that allows adipocytes uptake of those fatty acids included in the triglycerides transported by lipoproteins (chylomicrons and VLDL), no significant changes were observed (Figure 6). The gene expression of *dgat2*, the enzyme involved in the assembly of glycerol-phosphate and fatty acids into triglycerides, was also measured and it remained unchanged (Figure 6).

With regard to triglyceride mobilisation, specifically in the lipolytic pathway, the gene expression of *atgl* and *hsl*, two lipases implicated in triglyceride hydrolysis, remained unchanged (Figure 6). Taking into account that HSL serves as the rate-limiting enzyme in the catabolism of triglycerides [45], its activity was assessed by quantifying the levels of total and phosphorylated protein, in order to identify possible modifications at a posttranscriptional level. Figure 5 shows that both the peel and the pulp extracts increased the ratio of HSL-phosphorylated/total HSL. Considering that HSL is activated with phosphorylation, these results indicate that both extracts activated the lipolytic pathway.

Altogether, these results show that the main mechanism underlying the triglyceride-lowering effect of the whole fruit extract involves a reduction in de novo lipogenesis, attributed to the inhibition of ACC. Concerning the peel extract at 50 µg/mL, it not only inhibited de novo lipogenesis but also activated the lipolytic pathway by increasing the activity of the HSL. Finally, the pulp extract exhibited a dual effect by reducing de novo lipogenesis and the uptake of blood fatty acids, while simultaneously increasing lipolysis (Figure 7).

## 4. Conclusions

In summary, *O. stricta* var. *dillenii* fruit’s whole fruit, tissues (peel, and pulp), and by-products (bagasse) represent rich sources of betalains and phenolic compounds. Based on total betalain content, the pulp extract has emerged as the most abundant, demonstrating a high concentration of betanin and neobetanin. In terms of phenolic compounds, the peel extract shows the highest amount, primarily comprising piscidic acid and isorhamnetin glucosides. The by-product obtained from *O. stricta* var. *dillenii* industrialisation (bagasse) is the least rich in bioactive compounds. The studied *O. stricta* var. *dillenii* extracts also showed high antioxidant activity. Moreover, the extracts from whole fruit, peel, and pulp are effective in reducing triglyceride accumulation in murine mature adipocytes at doses of 50 μg/mL or higher. Interestingly, the mechanisms of action underlying these effects depend on the type of extract. All extracts significantly reduced the activity of the ACC enzyme, as assessed using the ACC-Phospho/total ACC ratio. Moreover, the pulp extract also reduced the expression of *fas*, both of which are involved in de novo lipogenesis. In addition, the lipolytic pathway was also activated in the cells treated with peel and pulp extracts, as evidenced by the higher HSL activity (HSL-Phospho/total HSL ratio). In the case of the pulp extract, a decrease in fatty acid uptake from the blood stream may also contribute to its triglyceride-lowering effect.

Among the four studied *Opuntia* extracts, the pulp extract showed the highest triglyceride-lowering effect (−37.4%), at a dose of 50 µg/mL, probably due to its high content of betalains. In light of this result, the pulp extract may be considered the most promising starting material to obtain betalains and phenolic compounds. Nevertheless, considering that the pulp fruit is an edible fraction appropriate for inclusion as foodstuff in the diet, other extracts, even if they exhibit a lower triglyceride-lowering effect, might be more suitable from a food market perspective for this purpose. The whole fruit extract induced a significant effect, leading to a 24.70% reduction in triglycerides. Consequently, fruits that may be considered unsuitable for market incorporation due to their appearance or small size, could prove to be interesting starting materials to produce functional extracts with potential anti-obesity effects. On the other hand, the peel extract, which reduced triglycerides by 23.60%, could be an interesting option within the context of sustainability and the circular economy, as it is derived from the food industry waste. In contrast, bagasse does not appear to be a useful extract.

## Figures and Tables

**Figure 1 nutrients-16-00499-f001:**
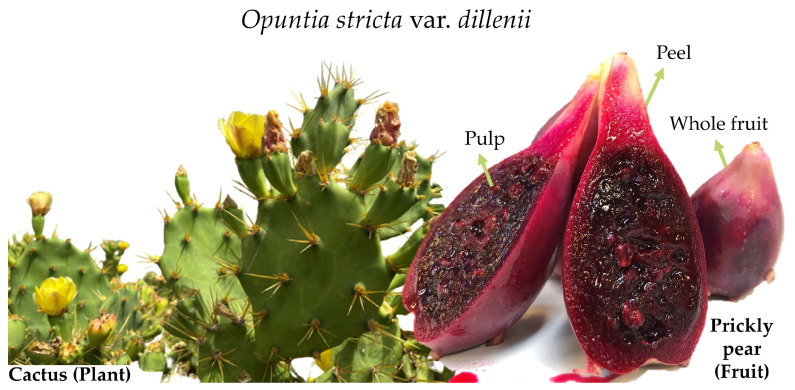
*Opuntia stricta* var. *dillenii* cactus and prickly pear from The Canary Islands, Spain.

**Figure 2 nutrients-16-00499-f002:**
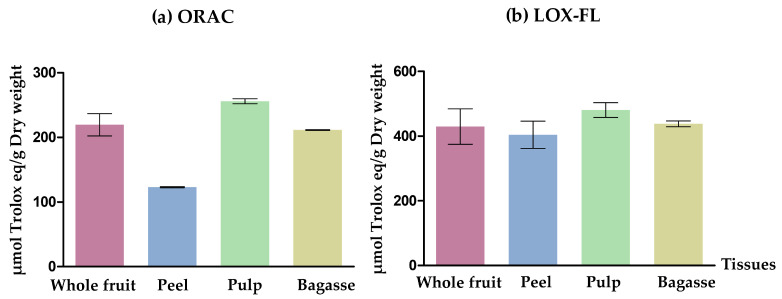
Antioxidant activity (**a**) ORAC and (**b**) LOX-FL of *Opuntia stricta* var. *Dillenii’s whole* fruit, tissues (peel and pulp) and by-product (bagasse) extracts.

**Figure 3 nutrients-16-00499-f003:**
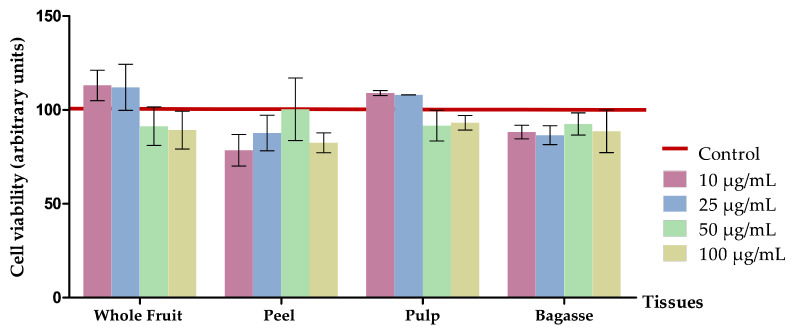
Effects of 10, 25, 50, and 100 µg/mL of extracts from *Opuntia stricta* var. *dillenii* whole fruit, peel, pulp, and bagasse on cell viability (%) of 3T3-L1 mature adipocytes treated for 24 h. Values are means ± SEM. Comparison between each extract dose and the control was analysed using Student’s *t*-test.

**Figure 4 nutrients-16-00499-f004:**
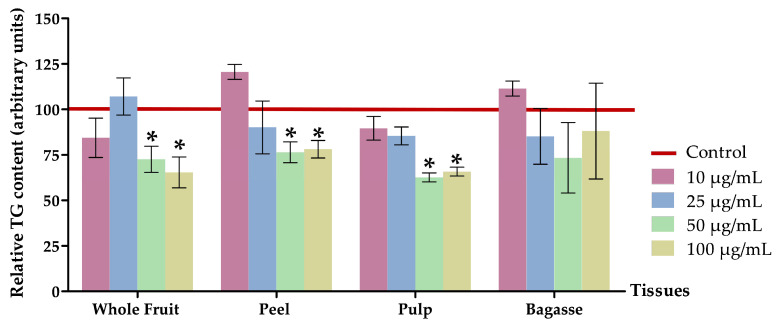
Effects of 10, 25, 50, and 100 µg/mL of extracts from *Opuntia stricta* var. *dillenii* whole fruit, peel, pulp and bagasse on triglycerides content (%) of 3T3-L1 mature adipocytes treated for 24 h. Values are means ± SEM. Comparison between each extract dose and the control was analysed using Student’s *t*-test. The asterisks (*) represent differences versus the controls (*p* < 0.05).

**Figure 5 nutrients-16-00499-f005:**
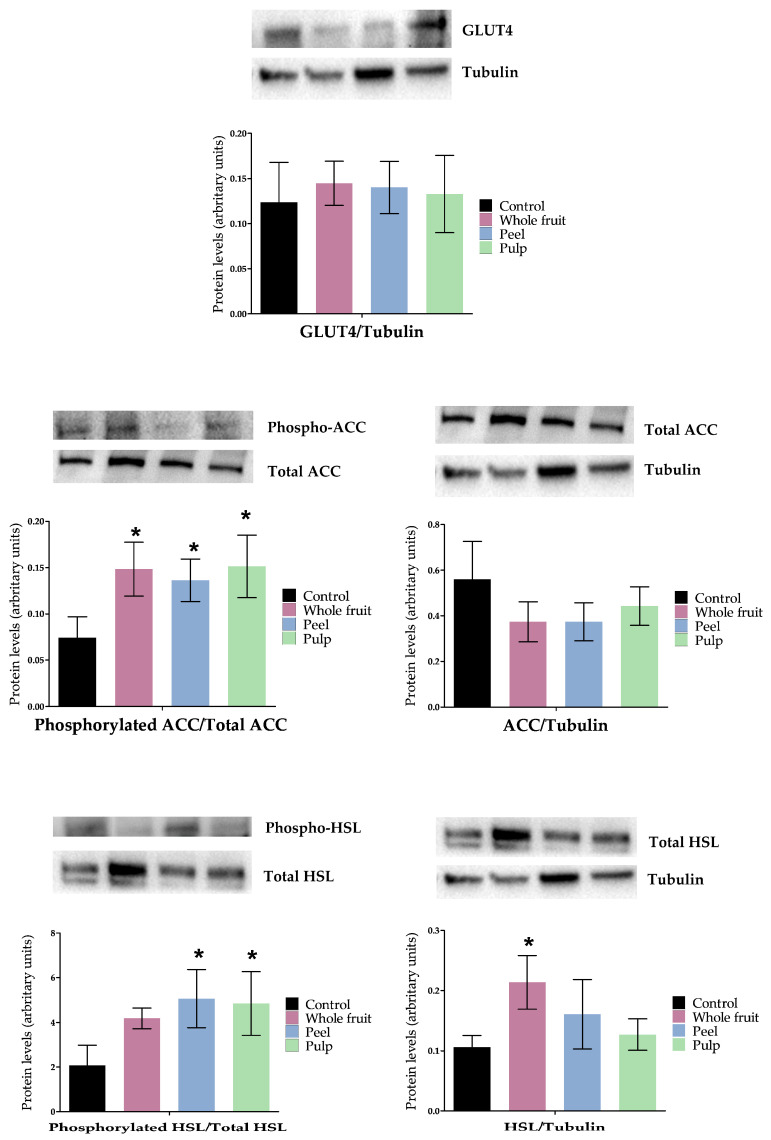
Effects of *Opuntia stricta* var. *dillenii*’s extracts on GLUT4, ACC, and HSL protein expressions and the ratios of phosphorylated-ACC/total phosphorylated-HSL/total HSL ratio in 3T3-L1 mature adipocytes treated for 24 h. Values are means ± SEM. Comparisons of extracts and the control for each gene expression were carried out using Student’s *t*-test. The asterisks (*) represent differences versus the controls (*p <* 0.05).

**Figure 6 nutrients-16-00499-f006:**
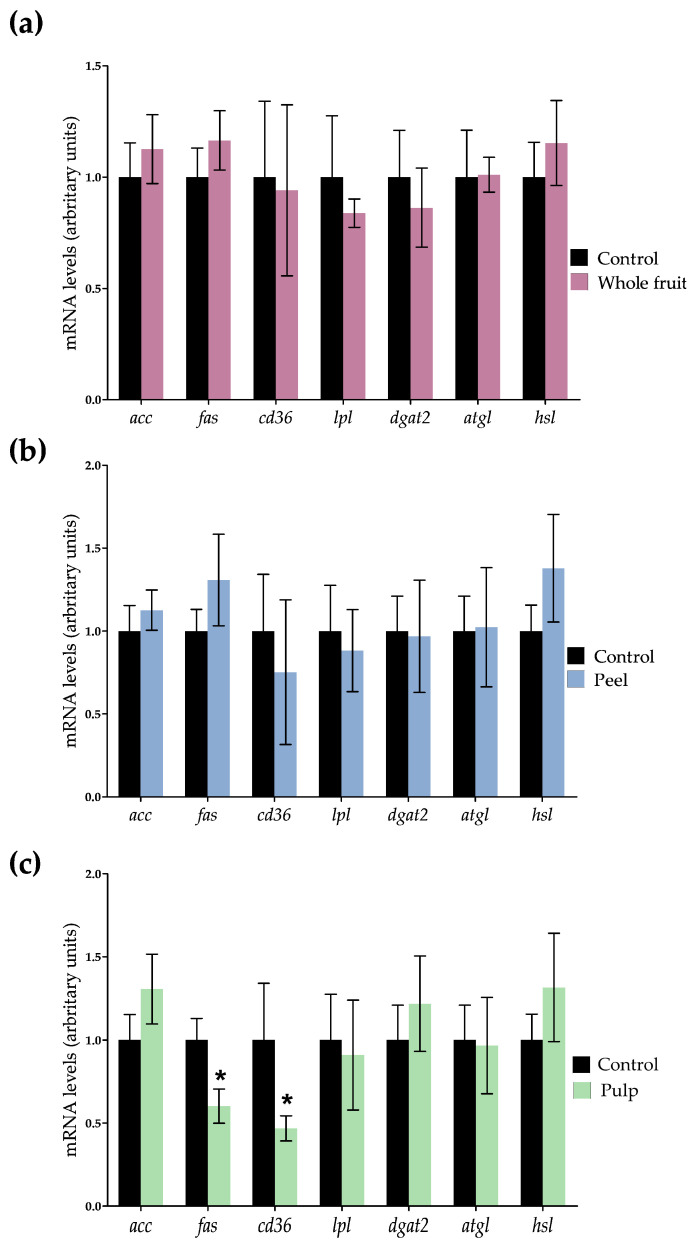
Effects of *Opuntia stricta* var. *dillenii* extracts from whole fruit at a dose of 50 µg/mL (**a**), from peel at a dose of 50 µg/mL (**b**) and from pulp at a dose of 50 µg/mL (**c**) on gene expression of *acc*, *fas*, *cd36*, *lpl*, *dgat2*, *atgl*, *hsl* in 3T3-L1 mature adipocytes treated for 24 h. Values are means ± SEM. Comparisons of each extract and the control for each gene expression was carried out using Student’s *t*-test. The asterisks (*) represent differences versus controls (*p <* 0.05).

**Figure 7 nutrients-16-00499-f007:**
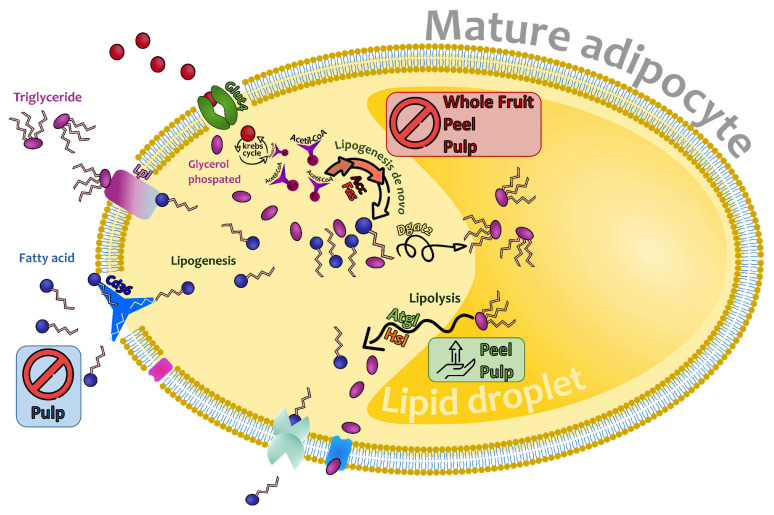
Overview of the gene expression of the lipogenesis and lipolysis of mature adipocytes and the main mechanism underlying the triglyceride-lowering effect of the whole fruit, peel, and pulp extracts from *Opuntia stricta* var. *dillenii.* GLUT4: Glucose transporter type 4; ACC: acetyl coenzyme A carboxylase; FAS: fatty acid synthase; CD36: cluster of differentiation 36; LPL: lipoprotein lipase; DGAT2: Diacylglycerol O-acyltransferase 2; ATGL: adipose triglyceride lipase; HSL: hormone-sensitive lipase.

**Table 1 nutrients-16-00499-t001:** Primer sequences for real-time PCR amplification of each studied gene.

SYBR Green RT-PCR			
Gene	Sense Primer	Antisense Primer	Annealing (°C)
*fas*	5′-AGC CCC TCA AGT GCA CAG T-3′	5′-TGC CAA TGT GTT TTC CCT G-3′	62.7
*cd36*	5′-GAT GAC GTG GCA AAG AAC AG-3′	5′-CAG TGA AGG CTC AAA GAT GG-3′	60.7
*lpl*	5′-CCT CTC TCC AGG GGA CAA GT-3′	5′-GAA GGC GGT CAA CTC TGG A-3′	60.0
*atgl*	5′-GAG CTT CGC GTC ACC ACC-3′	5′-CAC ATC TCT CGG AGG ACC A-3′	58.5
Taqman RT-PCR			
Gene	Assay ID		
*acc*	Mm01304285_m1		
*dgat2*	Mm00499536_m1		
*hls*	Mm00495359_m1		

*acc*: Acetyl coenzyme A carboxylase; *fas*: fatty acid synthase; *cd36*: cluster of differentiation 3; *lpl*: lipoprotein lipase; *dgat2*: diacylglycerol O-acyltransferase 2; *atgl*: adipose triglyceride lipase; *hsl*: hormone-sensitive lipase.

**Table 2 nutrients-16-00499-t002:** Bioactive compounds’ content (mg/g dry weight) of the major betalains and phenolic compounds in *Opuntia stricta* var. *dillenii*’s whole fruit, tissues (peel and pulp), and by-product (bagasse).

			*O. stricta* var. *dillenii*			
Peak *	Compounds	Family	Whole Fruit	Peel	Pulp	Bagasse
			mg/g Dry Weight			
1	Piscidic acid	Phenolic acid	1.64 ± 0.09 ^b^	2.33 ± 0.33 ^a^	0.62 ± 0.05 ^c^	1.54 ± 0.05 ^b^
2	Betanin	Betalain	2.97 ± 0.01 ^a^	2.99 ± 0.05 ^a^	2.91 ± 0.23 ^a^	0.84 ± 0.02 ^b^
3	Isobetanin	Betalain	1.85 ± 0.00 ^b^	1.65 ± 0.04 ^b^	2.28 ± 0.19 ^a^	0.77 ± 0.02 ^c^
4	Betanidin	Betalain	0.04 ± 0.00 ^a^	0.04 ± 0.00 ^a^	0.04 ± 0.01 ^a^	0.02 ± 0.00 ^b^
5	6′-O-sinapoyl-O-gompherin	Betalain	0.13 ± 0.00 ^b^	0.14 ± 0.00 ^a^	0.08 ± 0.00 ^c^	0.01 ± 0.00 ^d^
6	2′-O-apiosyl-4-O-phyllocactin	Betalain	1.29 ± 0.02 ^a^	1.22 ± 0.02 ^a^	1.61 ± 0.18 ^a^	0.58 ± 0.16 ^b^
7	5″-O-E-sinapoyl-2′-apyosil-phyllocactin	Betalain	3.14 ± 0.00 ^a^	3.23 ± 0.13 ^a^	2.60 ± 0.07 ^a^	n.d.
8	Neobetanin	Betalain	1.95 ± 0.02 ^b^	0.82 ± 0.00 ^d^	3.26 ± 0.05 ^a^	1.03 ± 0.07 ^c^
9	Quercetin-3-O-rhamnosyl-rutinoside (QG3)	Flavonoid	0.04 ± 0.00 ^b^	0.07 ± 0.00 ^a^	n.d.	0.02 ± 0.00 ^c^
10	Quercetin glycoside(QG1)—Quercetin hexosyl pentosyl rhamnoside	Flavonoid	0.04 ± 0.00 ^b^	0.08 ± 0.00 ^a^	n.d.	0.02 ± 0.00 ^c^
11	Quercetin glycoside(QG2)—Quercetin hexose pentoside	Flavonoid	0.02 ± 0.00 ^a^	0.02 ± 0.00 ^a^	n.d.	n.d.
12	Isorhamnetin glucosyl-rhamnosyl-rhamnoside(IG1)	Flavonoid	0.02 ± 0.00 ^a^	0.03 ± 0.00 ^a^	n.d.	0.01 ± 0.00 ^a^
13	Isorhamnetin glucosyl-rhamnosyl-pentoside(IG2)	Flavonoid	0.29 ± 0.00 ^b^	0.52 ± 0.02 ^a^	0.05 ± 0.00 ^d^	0.18 ± 0.01 ^c^
	Total major betalains		11.37 ± 0.02 ^b^	10.08 ± 0.03 ^c^	12.78 ± 0.48 ^a^	3.24 ± 0.27 ^d^
	Total major phenolic compounds (Pisicidic acid and flavonoids)		2.06 ± 0.09 ^b^	3.04 ± 0.02 ^a^	0.67 ± 0.05^d^	1.78 ± 0.06 ^c^
	Total major flavonoids		0.42 ± 0.00 ^b^	0.72 ± 0.02 ^a^	0.05 ± 0.00 ^d^	0.24 ± 0.01 ^c^

Results are expressed as mean ± standard deviation (n = 3). This is derived from obtaining a minimum of two independent extracts (n = 2) and conducting HPLC determinations on each occasion (n = 2). Superscript letters indicate statistically significant differences *p* ≤ 0.05) between different *O. stricta* var. *dillenii* tissues. n.d. no detected; * peak according to Appendix A.

## Data Availability

Data are contained within the article and Appendix A.

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
