# Peer review of "Anti-Obesity Effect of Different Opuntia stricta var. dillenii’s Prickly Pear Tissues and Industrial By-Product Extracts in 3T3-L1 Mature Adipocytes"

_nutrients, 2024, doi:10.3390/nu16040499_

Round 1

Reviewer 1 Report

Comments and Suggestions for Authors

This is an interesting work reporting the trygliceride lowering effects in adypocites of Opuntia stricta and could open the way to future nutraceuticals creation for pharmaceutical purposes. The manuscript should undergo major revisions

Introduction:

-Have you notices from the literature on using directly the purple OPD juice in therapeutical strategies? Or fresh juice of any other Opuntia species?

-pag 2 after the sentence 'On the one hand, these plants are a good source of numerous compounds, such as polyphenols, betalains and pectins, among others, that can induce beneficial effects on health' add some reference on the importance of Opuntia species in human health citing at least https://doi.org/10.3390/sym13061041 and https://doi.org/10.3390/horticulturae8020088

section 2:

Please check English in the sections relative to Methodology

2.3.1 'in Opuntia stricta var. dillenii 

(OPD) various tissues extracts' should probably be 'in  various tissues extracts of Opuntia stricta var. dillenii (OPD)'

Moreover, explain the abbreviation DAD in  HPLC-DAD. In general, all abbreviations should be explained at their first usage. Please check it.

2.3.1 explain already here why you selected the reported wavelenghts in the light of the literature for the detection of the different compounds families

2.4 Explain herein the reasons behind the choice of 3T3-L1 pre-adipocytes for the study.

2.4.2 no need of explaining PBS abbreviation again. Same in the following sections

section 3: provide a figure or scheme containing the structural representation of the main compounds identified in the OPD samples and mentioned in this section

3.1 any data on the levels of polyphenols, betalains,and other compounds in fresh juice of OPD?

figure 2: can we say that pulp is endowed with the highest antioxidant activity or this can not be stated considering error bars?

Figure 3. Judging on the red line position, peel and bagasse at low concentration lower even if slightly cell viability. please comment on it. Moreover, is there any reports on similar cytotoxicity studies on cancer cells?

3.2.2 Also here I wonder if fresh juice treatment could lower triglyceride content in adipocytes at a similar extent with respect to pulp....consider to perform such study may be in a future study. If there is something on it available from the scientific literature please report in the revised version of the manuscript.

end of pag 9, sect 3.2.2 inappropriate italics usage in the sentence  The authors declared....

figure 7: add a writing on the uptake and provide more detailed explaination on the content of the picture

section 5: remove bold sentences. Add a comparison of the triglyceride lowering effects of other Opuntias mentioning at least the work https://doi.org/10.1016/j.nutres.2011.05.002 on Opuntia humifusa

Comments on the Quality of English Language

English is generally good. However, language level should be improved in some ponts of the section 2

Author Response

Reviewer 2

Introduction:

-Have you notices from the literature on using directly the purple OPD juice in therapeutical strategies? Or fresh juice of any other Opuntia species?

Yes, talking about Opuntia stricta var. Dillenii (OPD), there is one article that evaluates the protective effect of fresh fruit juice of Opuntia dillenii Haw. (FJOD) on acetic acid-induced ulcerative colitis in rats (10.1080/19390211.2018.1470128 ). However, there are no articles about the effect of this variety on obesity.

-Pag 2 after the sentence 'On the one hand, these plants are a good source of numerous compounds, such as polyphenols, betalains and pectins, among others, that can induce beneficial effects on health' add some reference on the importance of Opuntia species in human health citing at least https://doi.org/10.3390/sym13061041 and https://doi.org/10.3390/horticulturae8020088

Following the reviewer's comment, the two proposed references have been included in this revised version, as well as two additional ones: 10.3390/plants11182333; 10.3390/antiox11122364 . References numbers: 8 to 11.

section 2:

Please check English in the sections relative to Methodology

2.3.1 'in Opuntia stricta var. dillenii (OPD) various tissues extracts' should probably be 'in  various tissues extracts of Opuntia stricta var. dillenii (OPD)'

We acknowledge the reviewer for this comment. The modification has been done in this revised version. 

Moreover, explain the abbreviation DAD in  HPLC-DAD. In general, all abbreviations should be explained at their first usage. Please check it.

We apologize for this mistake. The abbreviation has been explained in this revised version (page 3, line 104).

2.3.1 explain already here why you selected the reported wavelenghts in the light of the literature for the detection of the different compounds families

Following the reviewer's comment, this explanation has been added in this revised version (page 3, lines 112 to 115).

2.4 Explain herein the reasons behind the choice of 3T3-L1 pre-adipocytes for the study.

Murine 3T3-L1 adipocyte are a very well established cell line for determining the cellular mechanism involved in lipid accumulation in adipocytes and pre-adipocyte cell differentiation (adipogenesis). In fact, there are more than one thousand results on PUBMED when searching for the terms “adipogenesis and 3T3-L1” in the period of time of the last 5 years. Dufau et al. (2021) in their study about In vitro and ex vivo models of adipocytes concluded that the major methodological progress has been achieved since the pioneering development of the 3T3-L1mouse preadipocyte cell line (10.1152/ajpcell.00519.2020).

2.4.2 no need of explaining PBS abbreviation again. Same in the following sections

We acknowledge the reviewer for this comment. The modification has been carried out in this revised version.

section 3: provide a figure or scheme containing the structural representation of the main compounds identified in the OPD samples and mentioned in this section

Following the reviewer's suggestion, a new figure containing the structural representation of the main compounds identified in the OPD samples has been included as supplementary material in this revised version (Figure S2).

3.1 any data on the levels of polyphenols, betalains, and other compounds in fresh juice of OPD?

Yes, our previous research provides a complete characterisation of a raw juice from OPD (https://doi.org/10.3390/foods10071593).

figure 2: can we say that pulp is endowed with the highest antioxidant activity or this cannot be stated considering error bars?

In Table S2 at supplementary material, it can be observed that there are no statistically significant differences (P ≥ 0.05); P value are 0.189 and 0.089. Consequently, to conclude that pulp has the highest antioxidant activity is not possible.

Figure 3. Judging on the red line position, peel and bagasse at low concentration lower even if slightly cell viability. please comment on it. Moreover, is there any reports on similar cytotoxicity studies on cancer cells?

When comparing any treated cells to the controls, according to the statistical analysis performed, there are no significant differences in terms of cell viability. P value is 0.09 for bagasse at 10 µg/mL and P value is 0.08 for peel at 10 µg/mL.

Concerning the potential cytotoxic effect on cancer cells, we have not any information in the published studies. It is true that an anticancer effect has been reported for Opuntia extracts, but it is not due to cell toxicity is for the apoptosis (10.3389/fpls.2023.1236123;  10.1186/1475-2891-4-25)

3.2.2 Also here I wonder if fresh juice treatment could lower triglyceride content in adipocytes at a similar extent with respect to pulp....consider to perform such study may be in a future study. If there is something on it available from the scientific literature please report in the revised version of the manuscript.

We appreciate your suggestion and will keep it in mind for our next articles; a revision of the literature shows that there are no studies concerning this issue, so far.

end of pag 9, sect 3.2.2 inappropriate italics usage in the sentence  The authors declared....

This modification has been made in this revised version (page 10, lines 388-389).

figure 7: add a writing on the uptake and provide more detailed explaination on the content of the picture.

The picture has been prepared in order to help the understanding of the mature adipocyte biology and to summarize the effects of the extracts on the metabolic pathways of these cells. Due to fact that the pathways shown in the picture are explained in 3.2.3 section, we have not repeated this information in the footnote of the figure. Nevertheless, in order to make the figure clearer, we have added to the footnote the meaning of the protein abbreviations.

section 5: remove bold sentences. Add a comparison of the triglyceride lowering effects of other Opuntias mentioning at least the work https://doi.org/10.1016/j.nutres.2011.05.002 on Opuntia humifusa

We acknowledge the reviewer for this suggestion. Nevertheless, due to the fact that the study https://doi.org/10.1016/j.nutres.2011.05.002 has been carried out with a different Opuntia species, and using different parts of the plant, and consequently the bioactive compound composition is very different we have not considered to include this comparison in the publication text.

Reviewer 2 Report

Comments and Suggestions for Authors

1. LOX activity should be contracted in the context of a measurable in vitro effect with potentially health-promoting properties. Missing from the manuscript is a reference of the results of spectrofluorimetric studies on the future benefits of dietary supplementation of the proposed product. Also lacking is a control, inhibitor (drug) or reference sample (e.g., ORAC and anti-LOX activities for common dietary ingredients) in the context of the assayed activity. 

2. The identified compounds are only a percentage among flavonoids. Reference should be made to the other groups of flavonoids using extensive LC-MS methods.

Author Response

1. LOX activity should be contracted in the context of a measurable in vitro effect with potentially health-promoting properties. Missing from the manuscript is a reference of the results of spectrofluorimetric studies on the future benefits of dietary supplementation of the proposed product. Also lacking is a control, inhibitor (drug) or reference sample (e.g., ORAC and anti-LOX activities for common dietary ingredients) in the context of the assayed activity. 

The ORAC assay stands out as the most conventional method; however, LOX-FL antioxidant activity was also determined because this assay was very sensitive to hydrophilic, lipophilic, and phenolic antioxidant extracts, obtaining values at least 35 and 30 times higher than those by TEAC and ORAC methods, respectively (https://doi.org/10.1021/jf901509b). Soybean lipoxygenase-based methods are suitable alternatives for assessing the antioxidant capacity of betalains because they simultaneously detect the scavenging of physiological radical species, iron ion reducing and chelating activities, and inhibition of the pro-oxidant apoenzyme. Additionally, Gómez-Maqueo et al. [https://doi.org/10.1007/s11130-021-00914-7] proposed the LOX-FL method as the optimal choice based on its ability to better mirror the in vivo antioxidant capacity of betalains, substantiated by the correlation between the LOX-FL assay and betalain content. Soccio et al. (https://doi.org/10.3390/molecules23123244) also recommended the LOX-FL method to determine the antioxidant activity of a large variety of plant foods, which possesses a high ability to discriminate among different samples. The study concluded that LOX-FL demonstrated a superior real antioxidant capacity compared to other methods. This was particularly evident in the assessment of compounds, such as Lisosan G, allowing for both in vitro measurements of food extracts and ex vivo measurements of serum after food ingestion. The findings highlight the effectiveness of LOX-FL in accurately gauging the antioxidant potential of the tested compounds in various contexts (https://doi.org/10.1039/C6AY01002D).

The LOX-FL and ORAC antioxidant assays were determined using a Trolox dose-response curve. This helps to compare results among different experiments. Trolox serves as a standard in antioxidant assays, providing a consistent method for comparing the antioxidant capacity of different samples. This approach simplifies result interpretation and allows the expression of antioxidant activity in Trolox equivalents, facilitating comparisons across studies. It is important to note that the use of Trolox in ORAC assays is a standardized approach, and the results are expressed in Trolox equivalents rather than in terms of the specific antioxidant capacity of the sample. (https://doi.org/10.1016/B978-0-12-404738-9.00025-8).

These methods, not using a reference control, are widely used in the literature. Some examples of papers that have used them are the following: Sladanha et al. (2023) (https://doi.org/10.3390/metabo13020277), Cova et al. (2015) (https://doi.org/10.3109/09637486.2015.1088938), Santos Zea et al. (2011) (https://doi.org/10.1021/jf200944y), Sallam et al. (2022), (https://doi.org/10.3390/molecules27217568)

2. The identified compounds are only a percentage among flavonoids. Reference should be made to the other groups of flavonoids using extensive LC-MS methods.

The complete characterization of all individual betalains and phenolic compounds found in OPD extracts was previously reported by our research group in a study where the same extracts than in the present study were used. This is the reason that explains why in the present study we only report the most representative compounds in this cactus (Flavonoids included). You can find the details at https://doi.org/10.3390/foods10071593.

Reviewer 3 Report

Comments and Suggestions for Authors

The manuscript “ Anti-obesity effect of different Opuntia stricta var. dillenii’s prickly pear tissues and industrial by-product extracts in 3T3-L1 mature adipocytes” submited by Gomez-Lopes et al. Comprises the study of the triglyceride-lowering effect of green extracts of different parts of O. stricta. The work is important because it has been proven that natural extracts are capable of treating and curing various diseases. Furthermore, the species studied have already demonstrated that they are a source of natural products of biological importance. The study presented can add commercial and pharmacological value to the plant species studied.

The biological testing part seems to have been well prepared and is well described. Statistical analysis was conducted. However, some observations are presented below, as a suggestion to improve the text, in terms of clarity for readers.

In general, the genus name, Opuntia, must be abbreviated (O.) from the first time it is written in full in the text.

Also, please, verify if the author Eseberri has its name properly stated in the authorship line.

Section 2.1.

Knowing the influence of growth conditions on plant metabolism, it is necessary to be more specific in relation to where the pears are collected (lines 76-77).

In line 80, there is mention of Figure 1, but this figure does not show the difference between endocarp and exocarp, as suggested in the text.

Section 2.2

In line 96, the term "pure ethanol" gives the impression that the ethanol previously used to prepare the solvent mixture was not pure. Therefore, I suggest removing the word pure, or changing it to 100% ethanol.

Results and discussion section

The results and discussion are presented clearly. I suggest that the word "tissues" be removed from the text, for example, in Table 2, because whole fruit cannot be considered a tissue.

Please check the current IUPAC rules to see if the O in the nomenclature of several described compounds (e.g. 6'-O-sinapoyl-O-gompherin) must be in italics, to comply with international nomenclature standards.

The results are described and sufficiently discussed. The graphs are informative, although they could draw more reader attention if the bars were in different colors and not in shades of gray. But this does not take away the importance of the data presented.

The quality of the font in the graphics could be improved and should be checked. Ex: The word control is cut off in Figure 6C.

Conclusions

The conclusions somewhat repeat data already presented, but it is ok.

References

The references seem relevant and are up to date.

Author Response

In general, the genus name, Opuntia, must be abbreviated (O.) from the first time it is written in full in the text.

Following the reviewer's comment, this change has been carried out in the revised version.

Also, please, verify if the author Eseberri has its name properly stated in the authorship line.

We apologize for this mistake. We acknowledge the reviewer for this comment. The modification has been done in this revised version

Section 2.1.

Knowing the influence of growth conditions on plant metabolism, it is necessary to be more specific in relation to where the pears are collected (lines 76-77).

Following the reviewer's comment, this explanation has been added in this revised version (page 2, lines 76 to 77).

In line 80, there is mention of Figure 1, but this figure does not show the difference between endocarp and exocarp, as suggested in the text.

Our aim is to use Figure 1 as a tentative illustration of the fruit appearance, encompassing the peel, pulp, and the entire fruit. To prevent confusion, we have temporarily excluded the endocarp and exocarp from the text because we believed it was not clearly depicted. We apologize for this mistake, and the modification has been implemented in this revised version (Page 2, line 79).

Section 2.2

In line 96, the term "pure ethanol" gives the impression that the ethanol previously used to prepare the solvent mixture was not pure. Therefore, I suggest removing the word pure, or changing it to 100% ethanol.

Following the reviewer's comment, the modification has been made in this revised version (page 3, lines 96).

Results and discussion section

The results and discussion are presented clearly. I suggest that the word "tissues" be removed from the text, for example, in Table 2, because whole fruit cannot be considered a tissue.

Following the reviewer's comment, the modification has been made in this revised version.

Please check the current IUPAC rules to see if the O in the nomenclature of several described compounds (e.g. 6'-O-sinapoyl-O-gompherin) must be in italics, to comply with international nomenclature standards.

We apologize for this mistake and we acknowledge the reviewer for this comment. According to the revised Nomenclature of Organic Chemistry: IUPAC Recommendations and Preferred Names 2013, revised version (December 6, 2023, https://iupac.qmul.ac.uk/BlueBook/PDF/), indeed it is necessary to italicize the oxygen atom (-O-). The modification has been done in this revised version (Page 7, line 291, and Table 2, Table S1).

The results are described and sufficiently discussed. The graphs are informative, although they could draw more reader attention if the bars were in different colours and not in shades of gray. But this does not take away the importance of the data presented.

According to the reviewer's suggestion different colours have been used to built the figures.

The quality of the font in the graphics could be improved and should be checked. Ex: The word control is cut off in Figure 6C.

Following the reviewer's comment, the modification has been made in this revised version (Figure 6, page 12).

Round 2

Reviewer 1 Report

Comments and Suggestions for Authors

The paper can be accepted in the current  form 

Author Response

Thank you for your feedback